# Slot Blot- and Electrospray Ionization–Mass Spectrometry/Matrix-Assisted Laser Desorption/Ionization–Mass Spectrometry-Based Novel Analysis Methods for the Identification and Quantification of Advanced Glycation End-Products in the Urine

**DOI:** 10.3390/ijms25179632

**Published:** 2024-09-05

**Authors:** Takanobu Takata, Shinya Inoue, Kenshiro Kunii, Togen Masauji, Katsuhito Miyazawa

**Affiliations:** 1Division of Molecular and Genetic Biology, Department of Life Science, Medical Research Institute, Kanazawa Medical University, Uchinada 920-0293, Ishikawa, Japan; 2Department of Pharmacy, Kanazawa Medical University Hospital, Uchinada 920-0293, Ishikawa, Japan; masauji@kanazawa-med.ac.jp; 3Department of Urology, Kanazawa Medical University, Uchinada 920-0293, Ishikawa, Japan; s-inoue@kanazawa-med.ac.jp (S.I.); kenshiro@kanazawa-med.ac.jp (K.K.); 4Inoue Iin Clinic, Kusatsu 525-0034, Shiga, Japan

**Keywords:** lifestyle-related disease, advanced glycation end-products, urine, biomarker, membrane chromatography, slot blot, Takata’s lysis buffer, electrospray ionization-mass spectrometry, matrix-assisted laser desorption/ionization-mass spectrometry

## Abstract

Proteins, saccharides, and low molecular organic compounds in the blood, urine, and saliva could potentially serve as biomarkers for diseases related to diet, lifestyle, and the use of illegal drugs. Lifestyle-related diseases (LSRDs) such as diabetes mellitus (DM), non-alcoholic steatohepatitis, cardiovascular disease, hypertension, kidney disease, and osteoporosis could develop into life-threatening conditions. Therefore, there is an urgent need to develop biomarkers for their early diagnosis. Advanced glycation end-products (AGEs) are associated with LSRDs and may induce/promote LSRDs. The presence of AGEs in body fluids could represent a biomarker of LSRDs. Urine samples could potentially be used for detecting AGEs, as urine collection is convenient and non-invasive. However, the detection and identification of AGE-modified proteins in the urine could be challenging, as their concentrations in the urine might be extremely low. To address this issue, we propose a new analytical approach. This strategy employs a method previously introduced by us, which combines slot blotting, our unique lysis buffer named Takata’s lysis buffer, and a polyvinylidene difluoride membrane, in conjunction with electrospray ionization-mass spectrometry (ESI)/matrix-assisted laser desorption/ionization-mass spectrometry (MALDI-MS). This novel strategy could be used to detect AGE-modified proteins, AGE-modified peptides, and free-type AGEs in urine samples.

## 1. Introduction

Clinical examinations, typically part of a medical check-up, are conducted to diagnose physical conditions or diseases. Specimen examinations involve the use of samples such as body fluids [1].

Blood serves as the primary specimen for biochemical assessments and plays a crucial role in monitoring a wide range of diseases and general health status. Evaluations typically include blood counts, blood biochemistry, and the evaluation of coagulation system parameters [1,2,3,4,5].

Urine is another common biological specimen that is used for routine biochemical evaluation. Urine analysis dates back to 1956 when Dr. Alfred H. Free and his colleagues developed a technique to determine glucose levels in the urine [6]. The gold standard in urinalysis involves the semi-quantitative and qualitative dip-and-read methods, along with the microscopic examination of solid components in the urine, such as red and white blood cells and epithelial cells. Urine specimens can be used to evaluate vital biomarkers for kidney disorders, as various metabolites are excreted in the urine that can be used to assess kidney function [7,8,9,10,11,12,13,14,15]. Comprehensive urine analysis underscores its importance in diagnostics and disease monitoring. Saliva, in terms of clinical specimen collection, surpasses other sampling methods, such as blood, owing to its non-invasive nature, ease of collection, and cost effectiveness [16,17,18,19,20]. We believe that urine is a suitable sample to study the biomarkers of diseases or disorders given the high amounts of sample (e.g., 50–150 mL) that can be collected painlessly from the patient.

Diseases such as obesity [21,22], type 2 diabetes mellitus (T2DM) [23,24,25], non-alcoholic steatohepatitis (NASH) [26,27], cardiovascular diseases (CVD) [28,29], hypertension [30,31], kidney diseases (KDs) [32,33], and osteoporosis [34,35,36,37] are called lifestyle-related diseases (LSRDs), as their pathogenesis is closely related to lifestyle factors such as diet, physical activity, and sleep patterns. Previous studies have linked the development and progression of LSRDs to the presence of advanced glycation end-products (AGEs) in body fluids [38,39]. Notably, significant correlations have been observed between AGEs and conditions such as obesity [40], T2DM [41,42,43], NASH [44], CVD [45,46,47,48], KD [32], and osteoporosis [34,35,36,37].

AGEs are generated from saccharides and their metabolites/non-enzymic reaction products (subproducts) [38,39]. Low molecular weight AGEs (e.g., *N*^ε^-carboxymethyl-lysine (CML), *N*^ε^-carboxyethyl-lysine (CEL)), which were studied as early as 1912, were identified as metabolites/subproducts of saccharides that react with mono amino acids [38,49]. These AGEs were recently named free-type AGEs [49,50,51]. On the contrary, metabolites/subproducts of saccharides that react with amino acids in proteins are generally named AGE-modified proteins.

Various free-type AGEs have been detected and identified using nuclear magnetic resonance (NMR), gas chromatography–mass spectrometry (MS) (GC-MS), electrospray ionization–mass spectrometry (ESI-MS), and matrix-assisted laser desorption/ionization–MS (MALDI-MS), [49,50,51]. Furthermore, fluorescence measurements, Western blotting, slot blotting, and enzyme-linked immunosorbent assays (ELISA) can also be used to detect and quantify AGE-modified proteins.

This opinion article focuses on two aspects of AGEs. First, we examined a compound (i.e., glyceraldehyde) that is a metabolite or non-enzymatic reaction product capable of generating different types of AGEs, which we refer to as the crude AGE pattern. Second, we discussed how one or several free-type AGEs modify proteins in various forms. We categorized these patterns into the following four types—Type 1 and 2 diverse and Type 1 and 2 multiple AGE patterns [49,51]. Furthermore, these AGEs can be classified as intra- and extracellular AGEs [38,39,49,50,51]. For example, several AGE-modified proteins in skeletal muscle tissue and cardiomyocytes have been identified, quantified, and classified as intracellular AGEs. These AGE-modified proteins reportedly contribute to the dysfunction of skeletal muscle cells and cardiomyocytes [42,43,45,46,47]. In contrast, it is widely believed that intracellular AGEs can leak into body fluids, such as the blood [41,44,52,53], urine [34,35,36,37,40,48,54,55], and saliva [52], from cells in various organs. They are categorized as extracellular AGEs. This understanding of AGE categorization provides a comprehensive overview of their role in cellular function and disease progression.

AGEs in foods and beverages are called dietary AGEs and are classified as extracellular AGEs [56,57]. These extracellular AGEs can bind to the receptor for AGEs (RAGE) and toll-like receptor 4 (TLR4) in various cells and induce cytotoxicity and inflammation [58,59,60,61]. Since AGE levels in body fluid reflect dietary AGE intake levels, in addition to the generation and accumulation of intracellular AGEs in organs, it is challenging to use AGE levels in body fluids as biomarkers [49,51]. Therefore, there is an unmet need to develop novel technologies and strategies to identify and detect AGEs in biological samples and determine their functionality as biomarkers for various diseases and LSRDs.

Although the analyses of some AGEs such as CML and CEL [40,62] in the urine have been challenging, others, such as pentosidine, are routinely analyzed in the clinic [34,35,36,37,48,63,64,65]. The relationship between pentosidine levels in the urine and osteoporosis is well established, and pentosidine levels have been used as a biomarker of osteoporosis in the clinic [34,35,36,37]. However, the collection of proteins and peptides containing AGE-modified proteins and peptides from the urine is challenging because (i) their levels in the urine are very low and (ii) the processes used for collection, such as desalination, column chromatography, concentration, and precipitation of acetone [66,67,68,69], are extremely tedious. To overcome these challenges, we developed a membrane chromatography strategy using slot blotting [70,71,72]. Using this technique in tandem with Takata’s lysis buffer for cell and tissue lysis, we were able to quantify the intracellular AGE-modified proteins in cultured cells and tissues, albeit the structures of the targeted AGEs were not determined [49,50,73]. This slot blotting strategy could be used to collect and quantify AGE-modified proteins in the urine without the desalination, column chromatography, concentration, and acetone precipitation steps, which are involved in the conventional approach. This is because urine samples can be directly applied to polyvinylidene difluoride (PVDF) membranes and then blotted with appropriate anti-AGE antibodies [50,66,67,68,69,73].

Additionally, free-type AGEs and AGE-modified peptides can be identified using in-line ESI-MS/MALDI-MS, enabling structural analyses of the targeted AGEs [49,50]. However, the determination of free-type AGEs requires the acid hydrolysis of AGE-modified proteins and the collection of free-type AGEs, for which established methods are available [43,74,75,76]. For example, enzyme digestion of AGE-modified proteins followed by the collection, identification, and quantification of AGE-modified peptides have been reported in several studies [45,46,47,77]. AGE-modified proteins bound on the PVDF membrane could be subjected to acid hydrolysis to collect free-type AGEs similar to those in cell pellets. Additionally, AGE-modified proteins on the PVDF membrane could be subjected to “On-membrane digestion”, and the collected AGE-modified peptides could be identified and quantified using ESI-MS/MALDI-MS analysis [78,79,80]. These strategies for the successful identification and quantification of AGEs suggest that similar strategies using slot blot and ESI-MS/MALDI-MS could be used to develop a novel technique for the identification and quantification of AGEs in the urine. Furthermore, our novel method can avoid the need for pre-processing of urine required by existing methods and can perform multiple analyses of free-type of AGEs and AGE-modified proteins.

## 2. Clinical Urinalysis

The kidneys produce urine by filtering the blood. Urine produced by the kidneys is collected in the bladder through the ureters, and when a certain amount is accumulated, it is excreted through the urethra. The urine contains a variety of substances, including substances that are originally reabsorbed from the kidneys into the bloodstream, metabolites, and substances that are discharged into the urine due to diseases of the urinary tract. Since urine collection is easy and non-invasive, it is ethically acceptable to collect samples from healthy individuals.

Urine samples can be used for the development of biomarkers for several diseases. Currently, urinary biomarkers are used in the clinic to diagnose tubular disorders [8]. Urinary β2-microglobulin, α1-microglobulin, and N-acetyl-β-D-glycosaminidase (NAG) levels have been in clinical use for some time. Other novel biomarkers being developed for the diagnosis of renal injury include lipocalin 2 [11], which is rapidly induced in response to inflammation in the ascending limb of the Henle loop and collecting ducts during renal injury; the kidney injury molecule-1 (KIM-1) [12], which is expressed at the brush border of the proximal tubules that proliferate after renal injury; liver-1, which is induced in injured human proximal tubular cells; and liver-type fatty acid-binding protein (L-FABP) [13], whose expression is induced in damaged human proximal tubular cells.

In the field of oncology, urothelial carcinoma biomarkers include soluble substances or cells, exfoliated tumor cells, and changes in protein or DNA/RNA expression in the urine [7]. Cell-based biomarkers are used to detect tumor cells or their proteins or DNA/RNA in the urine. Bladder cancer 4 (*BLCA-4*), minichromosomal maintenance 5 (*MCM5*), human telomerase reverse transcriptase (*hTERT*), circulating tumor cells (*CTCs*), and cytokeratin 20 (*CK-20*) have been reported to be biomarkers of urothelial carcinoma. Urinary soluble tumor markers are used to detect soluble factors released by tumor cells and DNA and RNA in the extracellular endoplasmic reticulum. Typical examples include urinary bladder carcinoma antigen (UBC), CYFRA21-1, apolipoprotein-A1 (Apo-A1), interleukin-8 (IL-8), vascular endothelial growth factor (VEGF), C-C motif chemokine 18 (CCL18), HA or hyaluronidase (HAse), and soluble Fas (sFas) [14]. Circulating tumor DNA (ctDNA) and urinary tumor DNA (utDNA) in the plasma have been reported as potential biomarkers in liquid biopsy [15].

Urinary metabolomics can be used to analyze organic acids, amino acids, and derivatives such as pyrimidine and purine bases in the urine. Amino acid metabolism disorders, organic acid metabolism disorders, fatty acid metabolism disorders, and carbohydrate metabolism disorders, which are classified as inborn errors of metabolism, result in the excretion of altered levels of metabolites in the urine [9]. Therefore, changes in urine metabolites can be detected and used for their diagnosis. In addition, genetic disorders such as primary hyperoxaluria, cystinuria, adenine phosphoribosyltransferase (APRT) deficiency, 2,8-dihydroxyadeninuria, and Lesch–Nyhan syndrome can be diagnosed as the cause of recurrent urinary tract stone disease [9]. Recently, microRNAs in the urine were efficiently recovered using zinc oxide nanowires. Cancer patient-specific microRNAs are excreted in the urine not only in patients with urological cancers such as prostate and bladder cancer but in those with non-urological cancers such as lung, pancreatic, and hepatocellular cancers. Thus, it is possible to use urinalysis to screen for various cancers [10].

## 3. Lifestyle-Related Diseases

Lifestyle-related diseases (LSRDs) such as obesity [21,22], T2DM [23,24,25], non-alcoholic steatohepatitis (NASH) [26,27], cardiovascular diseases (CVD) [28,29], hypertension [30,31], kidney diseases (KD) [32,33], and osteoarthritis [34,35,36,37] pose significant health challenges globally, particularly in developed countries [39,49,50,51]. These conditions are closely associated with the consumption of excess nutrients, including sugars, lipids, and proteins. Historically, it is believed that LSRDs have affected humans for thousands of years, particularly among the noble and affluent classes who had access to excess nutrients [49,51]. With the advent of the twentieth century, the prevalence of LSRDs has increased in developed countries due to the widespread access to nutrient-rich diets.

The onset and progression of LSRDs are believed to be driven by the two following primary mechanisms: (i) the direct induction of organ dysfunction or damage accumulation (e.g., in the liver [26,27] and blood vessels [28,29]) due to the excess intake of sugars, lipids, and proteins, and (ii) the acceleration of organ senescence and functional decline (e.g., in the kidney [32,33] and bone [34,35,36,37]) due to nutrient overload. These mechanisms also involve the induction of AGEs [39,49,50,51]. Various LSRDs, including obesity [40], T2DM [41,42,43], NASH [44], CVD [45,46,47,48], KD [32], and osteoporosis [34,35,36,37], have been associated with AGEs.

## 4. Advanced Glycation End-Products

### 4.1. Origin of AGEs

In the classical categorization of AGEs, AGE-1, -2, -3, -4, -5, and -6 are named based on their respective origins [81]. AGE-1 is derived from glucose, while AGE-2, -3, -4, -5, and -6 are generated from glyceraldehyde, glycolaldehyde, methylglyoxal, glyoxal, and 3-deoxyglucosone, respectively (Figure 1) [81]. These compounds are metabolized and produced through the non-enzymic pathways from glucose, which is also the most upstream source for the generation of AGE-2, -3, -4, -5, and -6. Fructose-1-phosphate, generated from fructose via fructolysis, gives rise to glyceraldehyde, the origin of AGE-2 (Figure 1) [38,81].

The generation of AGEs typically occurs through pathways involving the Schiff base and Amadori products, collectively known as the Maillard reaction [38,81]. However, AGEs can also be generated through glucose and lipid oxidation, bypassing the Maillard reaction pathways. For instance, compounds such as glyceraldehyde [82], glycolaldehyde [83], methylglyoxal [84,85,86,87], and glyoxal [84,85,86,87] can react with proteins to generate intracellular AGEs. These AGEs can then leak into body fluids, such as blood, and be transported to other cells.

Hassel et al. reported that the ^13^C-labeling glyceraldehyde in mouse blood can cross the blood–brain barrier (BBB) and contribute to AGE generation in the brain [82]. This finding is significant, as it reveals a pathway for intracellular AGE generation in the brain despite the limited range of materials that can cross the BBB.

In contrast, AGE-2, -3, -4, -5, and -6 can be generated through saccharide and lipid oxidation [38,49]. The production of methylglyoxal and glyoxal from glycerol has been observed in the human body, food, medicines, and other industrial materials [88,89,90,91].

### 4.2. Free-Type AGEs

#### 4.2.1. Free-Type AGEs Containing One Amino Acid Residue

The detection and identification of the core structure of AGEs, specifically free-type AGEs, have advanced with the use of NMR, GC-MS, ESI-MS, and MALDI-MS [38,39,49,50,51]. Once the structure is identified, quantification can be performed using GC-MS, ESI-MS, and MALDI-MS. We believe that both ESI-MS and MALDI-MS have significantly advanced the identification and quantification of free-type AGEs and AGE-modified proteins in the 21st century [92,93]. Several AGEs, including CML [34,49], CEL [34,49], pyraline [49,94,95,96], 3-hydroxy-5-hydroxymethyl-pyridinium (GLAP) [49,77,97], *N*^ω^-carboxymethylarginine (CMA) [98,99], *N*^δ^-(5-hydro-5-methyl-4-imidazolone-2-yl)-ornithine (methylglyoxal-derived hydroimidazolone) (MG-H1) [45,46,49,74,77], glyoxal-derived hydroimidazolone (G-H1) [45,49,100,101], and argpyrimidine [45,46,49,77], have been reported to contain a lysine or arginine residue as the core structure (Figure 2). The free-type AGEs are shown in Figure 3. These AGEs have been detected in human and animal specimens as well as in cultured human cells [92,93,102,103]. With the advancement of technology, novel pathways for the generation of free-type AGEs have been reported. For instance, Tominaga et al. and Basakai et al. revealed that CML can be generated from both ribose and methylglyoxal [104,105]. Ban et al. reported a novel pathway wherein MG-H1 is generated from ribose without the intermediate step involving methylglyoxal [106]. In contrast, AGE-modified proteins are formed when these AGEs react with lysine or arginine in proteins [38,39,49,50,51].

#### 4.2.2. Free-Type AGEs Containing Two Amino Acid Residues

AGEs such as pentosidine [34,35,36,37], glucosepane [107,108,109], *N*^ε^-{2-[(5-amino-5-carboxypentyl)-amino]-2-oxoethyl}-lysine (GOLA) [110], glyoxal-derived imidazolium cross-link (GODIC) [111], methylglyoxal-derived imidazolium cross-link (MODIC) [111], and 3-doxyglucosone-derived imidazolium cross-link (DODIC) [111] contain two amino acid residues and have been detected in the human body (Figure 3). While collagen-bound pentosidine has been identified, free pentosidine has also been detected in blood and urine samples [34,35,36,37]. Novel AGEs containing two amino acid residues such as pyrrolopyridinium lysine dimer derived from glyceraldehyde (PPG) 1 and 2 [112], lysine-hydroxy-triosidine [113], and arginine-hydroxy-triosidine [113] have been produced in vitro. However, they have not been detected in the human body. Takeuchi et al. proposed a category of glyceraldehyde-derived AGEs known as “toxic AGEs (TAGE)” and hypothesized their structure. They predicted the existence of two types of TAGE, each containing two and three amino acid residues [114]. This hypothesis provides a valuable framework for predicting novel AGE structures, although it remains unproven. (Of note, Lee et al. named 1,3-di(*N*^ε^-lysiono)-4-methyl-imidazolium salt (methylglyoxal-lysine dimer, MOLD), an AGE generated from methylglyoxal as “toxic AGEs (TAGE)” [61]. However, these TAGEs are not identical.)

### 4.3. Crude, Diverse, and Multiple AGE Patterns

#### 4.3.1. Crude AGE Pattern

We have identified a pattern, termed the “Crude AGE Pattern”, wherein certain metabolites or non-enzymic reaction products can generate various structures of AGEs [49,51]. In a previous study, the human pancreatic ductal cell line (PANC-1) was treated with glyceraldehyde. This led to the identification and quantification of GLAP, MG-H1, and argpyrimidine-modified proteins using ESI-MS (Figure 4) [77]. The revelation that each of these AGEs is generated from glyceraldehyde validates that discovery.

#### 4.3.2. Diverse AGE Patterns

We have classified the different AGE patterns into two patterns, Type 1 and Type 2, based on how certain AGE structures modify specific proteins [49,51]. The Type 1 diverse AGE pattern involves certain AGE structures that can modify the same or different amino acids in one type of protein, but not one molecule of protein (Figure 5a). The Type 1 diverse AGE pattern can be further divided into three subcategories (Figure 5a). Type 1A—different types of AGEs modify the same amino acid residue in a single protein (one type of protein, but not one molecule of protein) [49,115]; Type 1B—same AGEs modify different amino acid residues in a single protein [49,99,115]; Type 1C—different AGEs modify different amino acid residues in a single protein [46,47,49,115]. These patterns can be validated using ESI- and MALDI-MS.

The Type 2 diverse AGE pattern involves a single type of AGE structure that modifies various types of proteins (Figure 5b). This pattern can be confirmed using Western blotting [42,47,73].

#### 4.3.3. Multiple AGE Pattern

We have categorized multiple AGE patterns into two patterns—Type 1 multiple AGE pattern and Type 2 multiple AGE pattern [49,51]. The Type 1 multiple AGE pattern involves certain AGEs that modify a single protein molecule but not a specific type of protein (one protein molecule but not one type of protein) (Figure 6a). To validate this pattern, peptides modified by more than two AGEs must be detected using ESI-MS or MALDI-MS [115]. The Type 2 multiple AGE pattern involves the modification of more than two proteins through intermolecular covalent bonds by an AGE structure (Figure 6b). However, proving this pattern with automatic ESI-MS or MALDI-MS analysis is challenging. The current software struggles to predict or input data when more than two peptides, originating from different proteins, combine with an AGE structure. Suppose AGEs combine with the same type of proteins (e.g., homodimer). In that case, their data generally are not predicted or inputted in the standard ESI-MS or MALDI-MS analysis software (e.g., Swiss-Prot database using the Masscot 2.1.0 search algorithm), making automatic identification unsuitable.

In contrast, if a specific type of anti-AGE antibody recognizes the AGE-modified proteins, the intermolecular covalent bond via AGE structures is not necessarily proven. This is because it may generate intramolecular covalent bonds, and other AGEs might produce intermolecular covalent bonds with some proteins.

#### 4.3.4. Influence of Crude, Diverse, and Multiple AGE Patterns on Cells

AGEs originating from the same precursor can generate various types of AGEs, leading to a “soup of AGEs” within cells (crude AGE pattern) [77]. Each AGE can induce various effects in its environment. Even within the same protein, differences in AGEs and modification sites can lead to different effects, such as dysfunction or activation (Type 1 diverse AGE pattern) [99,115]. When a single type of AGE modifies various proteins (Type 2 diverse AGE pattern) [42,47,73], the impact on each protein can vary, leading to dysfunction, activation, or aggregation. If one type of AGE can induce only single phenomenon (e.g., dysfunction) for all proteins and the same effect in various cells (e.g., apoptosis, necrosis), one type of AGE is able to be the universal key of the works of all proteins and living of cells, regardless of the molecular weight, folding, and other post-translational modifications such as glycosylation, phosphorylation, methylation, acetylation, and other types of AGEs [116,117]. However, this hypothesis has not yet been proven. In contrast, we predict that various types of AGEs can influence different proteins, and some types of AGEs may co-modify a single protein.

In the case where different AGEs modify a single protein (Type 1 multiple AGE pattern), this protein may exhibit activation of or reduction in function and gain resistance to protease and ubiquitin-mediated proteasome degradation by various types of AGEs, although this hypothesis had not yet been proven [116]. Different AGE modifications may influence the effect of each other within a single protein molecule, potentially leading to dysfunction or activation.

When a single AGE combines with more than two proteins (Type 2 multiple AGE pattern) [118], we hypothesize that there is the possibility of dysfunction occurring due to the structure inducing aggregation. Conversely, the function may be promoted or activated in this pattern.

## 5. AGEs and LSRDs

### 5.1. Intracellular AGEs and LSRDs

Various AGEs have been found to be associated with LSRDs [49,50,51]. Studies have shown that C57BI/6J and OB/OB mice when fed a high-fat, high-sugar diet exhibit the generation and accumulation of CML-modified proteins in their skeletal muscles [42]. A similar accumulation was also observed in C57BI/6J mice, which were fed a 60% fructose diet [43].

Mastrocola et al. suggested that CEL-modified proteins might induce lipogenesis in the skeletal muscles [43]. Their study indicated that the relationship between T2DM and sarcopenia might be mediated through intracellular AGEs. Papadaki et al. reported that MG-H1- and CEL-modified proteins were generated and accumulated in actin in the myofilaments in cardiomyocytes from patients with DM and heart failure [45]. They further suggested the possibility that the modifications of MG-H1, G-H1, methylglyoxal, and glyoxal in myosin and actin in myofilaments reduced calcium-activated force in patients with heart failure. This tendency is more common in patients with DM and heart failure compared with that in other patients [46].

Intracellular AGEs are the key to understanding the relationship between DM and heart failure. Various types of methylglyoxal-derived AGEs (MGO-AGEs, AGE-4) and 2-ammnonio-6-[4-(hydroxymethyl)-3-oxidopyridinium-1-yl]-hexanoate-lysine (4-hydroxymethyl-OP-lysine, hydroxymethyl-OP-lysine) which were generated from glycolaldehyde were generated and accumulated in ryanodine receptor type 2 (RyR2) in the cardiomyocytes in mice. Their amounts in older mice (≥20 months) were greater than those in RyR2 in young mice (4–6 months) [47]. This AGE-modification in RyR2 could induce its dysfunction, promoting the leakage of calcium ions.

### 5.2. Extracellular AGEs and LSRDs

#### 5.2.1. AGEs in Body Fluids

Numerous researchers, including our team, believe that various intracellularly generated AGEs are secreted or leaked into body fluids such as blood [41,44,53,119,120,121,122], saliva [52], and urine [34,35,36,37,48,54,55]. This process is generally responsible for the reorganization of AGEs. However, we propose that dietary AGEs exist in the body fluids and contribute to the generation of AGEs in these fluids (this point will be elaborated on in Section 5.2.3).

In 2022, Lyndk et al. detected total AGEs in serum using fluorescence and quantified the novel melibiose-derived AGEs (MAGEs) in serum using an ELISA [41]. Similarly, Litwinowicz et al. quantified MAGEs in plasma [44] in 2022. Pentosidine and CML were detected in the urine with ELISA and GC-MS [34,35,36,37,48,54,55]. In these studies, free-type pentosidine and CML were detected.

As RAGEs and TLR4 are expressed in various cells in some organs, the AGE-RAGE and AGE-TLR4 axes can induce inflammation [58,59,60,61,123]. Although AGE-modified proteins and peptides bind to these receptors, Lee et al. reported that free-type MOLD, which is generated from methylglyoxal, binds to the RAGE and induces reactive oxygen species (ROS) and phosphatidylinositol 3-kinase-protein kinase B and nuclear factor kappa B [61]. They reported that the free-type glyoxal-lysine dimer (GOLD)-RAGE axis induced oxidative damage and an inflammatory response [124].

Considering the presence of AGE precursors such as glyceraldehyde [82], glycolaldehyde [83], methylglyoxal [84,85,86,87], glyoxal [84,85,86,87] in blood, and methylglyoxal in urine [125], there is a possibility that AGEs are generated in body fluids. Therefore, the presence of glyceraldehyde, glycolaldehyde, methylglyoxal, and glyoxal in blood and urine requires further investigation.

#### 5.2.2. AGEs in the Extracellular Matrix

Collagen plays a crucial role in the extracellular matrix of bone tissues [34,35,36,37,118]. The structure of pentosidine, which can bind two amino acid residues (as shown in Figure 3), has the potential to create non-physiological bridges in collagen. This alteration can reduce collagen strength and flexibility, potentially leading to conditions such as osteoporosis and related diseases such as osteoarthritis [34,35,36,37,118].

#### 5.2.3. Dietary AGEs

Saccharides, such as glucose and fructose, are abundant in many foods and beverages and are often produced and cooked with heat in manufacturing processes [56,57,126,127]. This process promotes the Maillard reaction, which leads to the generation of various AGEs. Wada reported that a higher intake of CML was significantly associated with an increased risk of liver cancer, according to the ‘Takayama Study’ [56].

Lin et al. analyzed the presence of CML, CEL, MG-H1, G-H1, argpyrimidine, MOLD, and GOLD in foods purchased from a local supermarket in Taipei [126]. Chen et al. focused on dietary CML, CEL, and MG-H1 and explored the relationships between the intake of AGEs and skin AGEs [127].

These dietary AGEs can bind to RAGEs and TLR4 and induce cytotoxicity [57,123]. Jaggupilli et al. focused on the bitter taste receptors (T2Rs) rather than RAGE/TLR4 and found that CML and GOLD were able to bind to T2Rs. Their investigation is expected to advance research on the dietary AGE-T2Rs axis [128].

#### 5.2.4. Fluid AGEs as a Biomarker

Numerous researchers have faced the challenge of detecting and quantifying individual AGEs in serum [41,53,119], plasma [44,120,121,122], saliva [52], and urine [34,35,36,37,48,54,55] as biomarkers of LSRD. The quantification of pentosidine in urine and skin as a biomarker of LSRDs is discussed later (please see Section 5.2.5). Litwinowicz et al. determined the structure of MAGEs and found that MAGEs in plasma could serve as a beneficial biomarker for alcoholic liver disease and stages of alcoholic hepatitis [44]. In a fluorescence quantification analysis, Lyndk et al. quantified total AGEs in serum to assess T2DM [41], while Pinto et al. quantified total AGEs in plasma to assess carotid atherosclerotic plaques [122].

Jung et al. reported an increase in salivary AGEs using an anti-AGE antibody (Cat. no. 6D12; TransGenic, Kobe, Japan) in older Sprague-Dawley rats [52]. Baskai et al. developed and modified the analysis of CML and CEL in urine using GC-MS to establish a biomarker. Wada et al. analyzed CML in the blood and measured dietary CML, suggesting that CML in the blood could reflect nutritional habits [56]. Takeuchi et al. proposed that serum TAGEs were a beneficial biomarker for T2DM, cardiovascular disease (CVD), non-alcoholic steatohepatitis (NASH), Alzheimer’s disease, infertility, and cancer though the structure of TAGEs remains unproved [81].

#### 5.2.5. Pentosidine in the Urine and Skin as a Biomarker for Aging, Osteoporosis, and CVD

Pentosidine in urine has been a significant focus as a biomarker [34,35,36,37,48,63,64,65,118]. Yoshihara et al. revealed the relationships between aging and urinary pentosidine [63]. Subsequent studies have shown an increase in urinary pentosidine in older patients with T2DM or inflammatory bowel disease [64,65]. We propose that urinary pentosidine is a particularly beneficial biomarker for osteoporosis and associated diseases [34,35,36,37,118]. Furthermore, the p-nitrosidine in the skin could serve as a biomarker for osteoporosis-induced fragility fractures as well as urine. This is supported by the proven relationships between bone pentosidine and urine pentosidine (r = 0.355, *p* = 0.026), skin pentosidine and urine pentosidine (r = 0.409, *p* = 0.002), and bone pentosidine and skin pentosidine (r = 0.422, *p* < 0.001) [35].

A relationship was identified between skin pentosidine/collagen and cartilage pentosidine (r = 0.473, *p* = 0.055) and mean urine pentosidine/creatinine and skin pentosidine/collagen (r = 0.285, *p* < 0.001) was identified in patients with osteoarthritis [36]. Given that urine pentosidine levels reflect the pathological conditions of osteoporosis, fragility fractures, and osteoarthritis, they could serve as clinical stage biomarkers [34,35,36,37,118]. Furthermore, urine pentosidine could also be associated with CVD. Kashiwabara et al. developed a novel ELISA method to quantify the free-type pentosidine in urine for the assessment of CVD risk factors [48].

## 6. Previous Methods Used for the Identification and Quantification of AGEs in the Urine

We believe that the identified and quantified AGEs in the urine in previous studies were actually pentosidine [34,35,36,37,48,64,65], CML [40,55], and CEL [40,55]. For the analysis of pentosidine, high-performance liquid chromatography (HPLC), MS methods such as GC-MS, and ELISA were adopted. Several studies have used the acid hydrolysis of pentosidine-modified proteins to collect free-type pentosidine, followed by analyses via HPLC [34,35,36,37,48,64]. Pentosidine has been detected based on the excitation wavelength of 335 nm and emission wavelength of 385 nm with HPCL analysis [34,48], whereas other studies have detected pentosidine based on the excitation wavelength of 328 nm and emission wavelength of 378 nm, and the collected and purified samples were identified with MS such as GC-MS [36,37]. Kashiwabara et al. developed an ELISA capable of quantifying free-type pentosidine and correlated the HPLC and ELISA findings for urinary pentosidine (r = 0.815) [48]. Kato et al. developed a competitive ELISA to quantify pentosidine-modified proteins, as urine samples could not be subjected to acid hydrolysis [65]. Baskai et al. identified and quantified free-type CML and CEL in the urine [40,55]. The urine was allowed to evaporate under a stream of nitrogen, and the residues were reconstituted in aliquots of a 2 M solution of hydrochloride/methanol to prepare the methyl ester derivative [40,55]. As the range of molecular weight for detection via GC-MS analysis is 100–1000 kDa, CML and CEL can be easily identified [40,55]. Furthermore, each compound had a specific retention time, which allowed the quantification of free-type CML and CEL through GC-MS.

We concluded that most investigations of the presence of AGEs in the urine involved (i) free-type AGEs obtained following the acid hydrolysis of AGE-modified proteins and (ii) free-type AGEs present in the urine.

## 7. Novel Analysis Method Combining Slot Blot and ESI-MS/MALDI-MS for the Identification and Quantification of AGEs in Urine

### 7.1. Concept and Novelty

The novelty and concept of this method are based on the following two main factors: (i) the method provides a solution to the challenge of collecting AGE-modified proteins in urine and (ii) the method facilitates the simultaneous analysis of AGE-modified proteins and free-type AGEs from the same samples. To achieve this goal, the slot blot and ESI-MS/MALDI-MS analyses should be integrated.

### 7.2. Steps Involved in the Novel Analysis Method for AGEs in Urine

The flow chart in Figure 7 summarizes the steps involved in the novel analysis method. The urine of patients or model animals is collected. The process begins with the collection of urine from patients or model animals. This urine is then centrifuged to remove cells that have sloughed off the urethra and entered the urine, as well as any crystals [66,67,68,69]. In this step, researchers can use existing methods to concentrate urine [66,67,68,69]; however, if the concentration obtained is lower, the collected supernatants can be repeatedly deposited onto the PVDF membrane [117]. In contrast, acetone precipitation is not to be performed using our novel method as this step is performed in existing methods to collect proteins rather than to concentrate the sample [69]. The collected supernatant is then deposited into the lanes of the slot blot apparatus equipped with a PVDF membrane. This step should be performed repeatedly because this will allow the concentration of the sample to further increase that is then deposited onto the PVDF membrane. After a sufficient amount of concentrated supernatant is deposited, AGE-modified proteins are bound to the PVDF membrane [50,73]. Next, Takata’s lysis buffer is added to the PVDF membrane, causing the AGE-modified proteins to degenerate through carbamoylation and bind to the membrane [50,117]. After cutting the PVDF membrane, each sample can be analyzed using the following three methods: Method 1: on-membrane acid hydrolysis is performed, and the collected free-type AGEs are analyzed with ESI-MS/MALDI-MS [43,74,75,76]; Method 2: on-membrane enzymatic digestion is performed, and the collected AGE-modified peptides are analyzed with ESI-MS/MALDI-MS [45,46,47,77]; Method 3: AGE-modified proteins on the membrane are analyzed with a slot blot using anti-AGE antibodies [50,73]. Each of these methods is described in detail below (please see Section 7.3, Section 7.4 and Section 7.5).

### 7.3. Overcoming the Challenges Associated with the Collection of AGE-Modified Proteins in Urine

AGE-modified proteins, including methyl, acetyl, phosphorylated, and glycosylated proteins, are not abnormal substances, although their function may be affected by AGE modification [117]. Therefore, their collection should be performed like that of general proteins. However, the concentration of proteins and peptides in urine is typically low, requiring processes such as concentration, column chromatography desalination, and precipitation of acetone [66,67,68,69].

Centrifugation aims to remove cells (e.g., squamous epithelial cells) and crystals containing calcium oxalate and phosphate and to collect large amounts of AGE-modified proteins from the concentrated supernatant. For instance, Yang et al. performed the ultrafiltration of a urine sample (12,000× *g*, cut-off, 10 kDa) [67]; Bellei et al. centrifuged 10 mL of urine (800× *g*, 4 °C) and collected the supernatant, and 4 mL of the supernatant was desalted and concentrated to a final volume of 100–120 μL using desalting spin columns (cut-off, 3 kDa) [68]; Zhang et al. centrifuged urine (12,000*× g*, 4 °C) to remove pellets such as cells and crystals, and then added four volumes of acetone to the supernatant [69]. However, this step can be challenging.

Although several strategies have been developed for the initial processing of the urine sample, concentration and desalination remain important steps. To address this, we used membrane chromatography [70,71,72]. Membrane chromatography can be categorized into the following two types: (i) samples flow perpendicular to the membrane and (ii) samples flow horizontally across the membrane (Figure 8). Samples can be bound onto the membrane based on factors such as molecular weight and affinity.

In our method, AGE-modified proteins can be bound onto the PVDF membrane (pore size; 0.45 μm). Low molecular weight compounds such as calcium, potassium, and chloride ions are removed (Figure 8a). We believe that the AGE-modified proteins in urine supernatant, after the removal of cells and crystals by centrifugation, can be collected by membrane chromatography, and slot blotting is suitable for this purpose [50,73].

### 7.4. Slot Blotting and Analysis

#### 7.4.1. Binding of AGE-Modified Proteins in Urine onto PVDF Membrane

We believe that slot blotting is an appropriate method to collect AGE-modified proteins in urine. Although various membrane materials, such as nitrocellulose [129,130,131,132] and PVDF [73,133], can be used for general AGE slot blotting in cell lysates, tissue lysates, and serum, we chose the PVDF membrane due to its superior strength. The samples are drawn using an aspirator (e.g., water aspirator) [50,117], and we believe that a strong membrane is suitable for this process (Figure 9). The general slot blotting apparatus, for example, the Bio-dot SF microfiltration apparatus (Bio-Rad, Hercules, CA, USA), incorporates a membrane (9 × 12 cm) divided into 48 lanes (Figure 10) [50,117]. The maximum volume for each lane is 500 μL, but the recommended volume of the sample solution to apply in the lane of the slot blot apparatus is 100–250 μL. If necessary, the sample solution can be repeatedly applied to the PVDF membrane [117].

#### 7.4.2. Carbamoylation of AGE-Modified Proteins onto the PVDF Membrane

We believe that carbamoylation of AGE-modified proteins enhances their adherence to the surface of the PVDF membrane [50,117]. This strategy is based on our previous study, in which we quantified the intracellular AGEs of cultured cells and tissues using a new slot blot method [50,73].

The core technology of this method is a new lysis buffer, recently named Takata lysis buffer, which contains tris-(hydroxymethyl)-aminomethane (Tris base), urea, thiourea, and (3-[(3-cholamidopropyl)-dimethyl-ammonio]-1-propane sulfonate) (CHAPS) [49,134]. We have used Takata’s lysis buffer and the modified Takata’s lysis buffer to quantify intracellular AGEs in cultured cells and tissues and have reported twelve references from 2017 to 2022 [50]. The detailed protocol was published as a separate article [134].

Although we believe the PVDF membrane is suitable for protein binding, commercial lysis buffers containing Triton-X may inhibit protein adherence [49,117,134]. Therefore, we developed a suitable lysis buffer for use with the PVDF membrane. Takata’s lysis buffer was developed based on the lysis buffers selected for proteomic analyses based on two-dimensional gel electrophoresis (2DE) [135,136,137,138]. We provide information on Takata’s lysis buffer and predicted carbamoylation mechanisms in Table 1 and Table 2 and Figure 11. Before Takata et al. reported the quantification of intracellular AGEs with the novel slot blot analysis using Takata’s lysis buffer [50,73], no study had reported the quantification of AGEs, with an accompanying statistical analysis, in the lysates of cultured cells/tissues, plasma, serum, saliva, urine, or other body fluid using the slot blot method with PVDF membranes. Although other groups have provided quantitative data indicating the amounts of intracellular AGEs, error bars were not included in their published graphs of the data [50]. We believe that the novel lysis buffer proposed by Takata will enhance the binding and probing of proteins, including AGE-modified proteins, onto PVDF membranes, which will allow statistical analysis of the data.

We prepared solution A (pre-Takata’s lysis buffer) and B in the first step; solution C (Takata’s lysis buffer) was prepared by mixing A and B [49,50,134]. We used the complete tablets EDTA-Free, EASY pack (Cat. no.: 04-693-132-001; Roche, Penzberg, Bavaria, Germany) as the protease inhibitor [49,50,117,134]. Since lysis buffers generally contain a protease inhibitor, and 4 *v*/*v* (%) Solution B should be included in Solution D as the normal standard method (one protease inhibitor cocktail tablet/50 mL of Solution D, based on the protocol provided by Roche), Solution D was not named individually. It is the modified Takata’s lysis buffer (or modified Solution A)

For the quantification of intracellular AGEs using the novel slot blot method, we prepared Takata’s lysis buffer based on eight previous studies. In comparison, Solution D was prepared based on four previous studies (as detailed in Table 2). Although no studies have directly compared the use of each solution in the same cells, we deemed both Solution C and Solution D suitable for the preparation of the cell lysate. Consequently, we utilized both solutions to perform the novel slot blot analysis.

We hypothesize that our lysis buffer enhances the protein binding on the PVDF membrane surface. As noted by McCarthy et al. and Herbert [150,151], urea, thiourea, and CHAPS can denature proteins by acting as chaotropic reagents and surfactants. These reagents disrupt hydrogen, leading to protein unfolding and the exposure of hydrophobic amino acid residues to the solution. CHAPS is combined with urea and thiourea to coat hydrophobic residues and improve solubility. The combinations of thiourea and urea are widely used to exploit the superior denaturing ability of thiourea [150].

Furthermore, urea may play a crucial role in inhibiting protein probes on the PVDF membrane. Urea reacts with ammonium and cyanate, with cyanate being particularly effective at producing isocyanic acid. Subsequently, this can react with *N*-terminal amino groups, as well as lysine, arginine, and cysteine residues in proteins, producing carbamylated proteins (Figure 11) [151]. Given that carbonyl (C=O) and amide (N–H) groups of the protein react with the PVDF membrane [152], carbamoylation may promote protein adhesion. Furthermore, Tris has been used to stabilize the pH range of cell lysates at values of 8.5–8.8 [117].
Figure 11Mechanism underlying the formation of carbamylated protein with urea [150,151]. (**a**) The reaction path of ammonium and isocyanate from urea. Two-way arrow indicates the reversible reaction. (**b**) Isocyanic acid attack on N-terminal lysine, arginine, and cysteine residues in protein. Closed blue circles indicate the amino acids. The black numbers represent the residue number in protein. Arrow indicates the reaction of carbamylation.
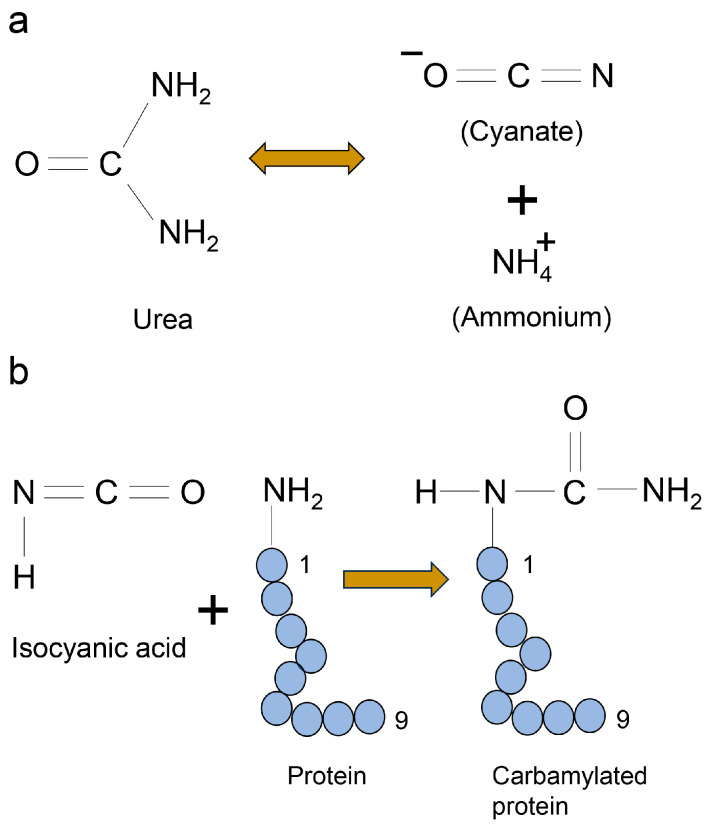



We believe that carbamoylation is necessary for adhering the proteins onto the PVDF membrane and will aid in the subsequent steps. This is because the binding strength of AGE-modified proteins against the PVDF membrane in slot blotting may be weaker than that in Western blotting. Although we prepared lysates from cultured cells and tissues using Takata’s lysis buffer (or modified Takata’s lysis buffer) [50,73,134,139,140,141,142,143,144,145,146,147,148,149], we propose that a certain volume (e.g., 25–100 μL) of this lysis buffer should be added at room temperature for a specific duration and removed with an aspirator, and the PVDF membrane should be washed with water. During the quantification of intracellular AGEs using our slot blot analysis, standard AGE-modified bovine serum albumin with Takata’s lysis buffer is added at room temperature for 5 min [134].

#### 7.4.3. Cutting the PVDF Membrane

After the AGE-modified proteins are bound to the PVDF membrane, these samples can be analyzed using three methods. The PVDF membrane can be cut into three pieces, and all three analyses can be performed simultaneously. Alternatively, only the required analysis may be performed.

#### 7.4.4. Slot Blot Analysis for AGE-Modified Proteins

In this step, AGE-modified proteins bound to the PVDF membrane can be detected and quantified using anti-AGE antibodies, though the structure of AGEs remains unclear [50,133,153]. Although this method cannot be used to determine the diverse and multiple AGE patterns (Figure 5 and Figure 6), slot blot analysis is beneficial for the detection and quantification of AGE and AGE-modified proteins.

### 7.5. ESI-/MALDI-MS Analysis of the AGE-Modified Proteins on the PVDF Membrane

#### 7.5.1. ESI-/MALDI-MS in AGE Research and Challenges Associated with Their Use for Detecting AGE-Modified Proteins

ESI-MS and MALDI-MS have been used to identify and quantify various proteins, including those modified by AGEs, contributing significantly to the development of proteomics research [154,155]. Several studies have identified different peptides after enzymatic digestion using MS and the relevant database. These peptides require the reduction of the disulfide bond and alkylation of the thiol group for both ESI-MS and MALDI-MS analyses [154,155]. However, the ionization step differs. ESI-MS can perform ionization without the need for additional reagents, whereas the MALDI-MS-subjected samples must be mixed with a matrix reagent for ionization. Furthermore, multiple charged peptide ions are easily generated in ESI-MS compared to those in MALDI-MS. On the contrary, single charged peptide ions are easily generated in MALDI-MS [45,46,47,154,155]. ESI-MS is typically coupled with HPLC to form HPLC-ESI-MS. This combination allows for the separation of compounds before ionization, giving them an advantage over MALDI-MS. As the retention times of peptides (AGE-modified peptide) are specific for each peptide, their quantification can be performed easily based on the HPLC-ESI-MS analysis [154,155]. However, MALDI-MS has also recently been integrated with HPLC [154,155]. The identification of novel AGEs (free-type AGEs) has been advanced through ESI-MS and MALDI-MS analyses [44,156]. However, these methods are not suitable for identifying or quantifying whole proteins, and AGE-modified proteins cannot be analyzed without acid hydrolysis or enzymatic digestion (Figure 7).

#### 7.5.2. ESI-/MALDI-MS-Based Analysis of Free-Type AGEs after On-Membrane Acid Hydrolysis

The technology for collecting free-type AGEs from cell lysate, serum, and food involves acid hydrolysis [16,74,75,76,102,126,157,158]. Acid hydrolysis is typically performed using 6 or 12 N hydrochloric acid. On the basis of previous methods, AGE-modified proteins on the PVDF membrane can be subjected to acid hydrolysis. Subsequently, free AGEs can be analyzed using ESI-MS or MALDI-MS (Figure 12).

GC-MS, along with ESI-MS and MALDI-MS, have contributed to the analysis of free-type AGEs, allowing the identification and quantification of various AGEs in human and animal fluids, such as urine and foods [43,54,55,159,160]. Free-type AGEs, after acid hydrolysis on a membrane, can be analyzed using GC-MS. However, these free-type AGEs must be derivatized to esters to increase their volatility [43,54,55,159,160]. Therefore, ESI- or MALDI-MS is often preferred.

#### 7.5.3. ESI-/MALDI-MS Analysis of AGE-Modified Peptides after On-Membrane Enzymatic Digestion

In proteomics investigations, “in-gel enzymatic digestion” is beneficial because the proteins in the gel cannot be collected into a tube. However, researchers can collect peptides from the gel and analyze them using ESI-MS or MALDI-MS. “In-gel enzymatic digestion” has been widely used for decades [161,162,163,164]. In contrast, “on-membrane enzymatic digestion” has gained attention and is performed to analyze proteins [78,79,80]. We focused on this technology because AGE-modified proteins on the PVDF membrane can be analyzed with ESI-MS or MALDI-MS. Since the identification and quantification of AGE-modified peptides in cell and tissue lysates and body fluids (e.g., serum) have developed and progressed [45,46,47,77,79,115], we believe that AGE-modified peptides collected after “on-membrane enzymatic digestion” can be analyzed. AGE-modified proteins are digested with enzymes such as trypsin, and some AGE-modified peptides can be collected for analysis with ESI-MS or MALDI-MS (Figure 13).

These analyses can be used to demonstrate Type 1 diverse and Type 1 multiple patterns (Figure 5a and Figure 6a). If various AGEs are found to modify the same amino acid sequence, the Type 1 diverse pattern can be confirmed (Figure 5a). However, characterizing Type 1 multiple patterns is challenging, even with ESI-MS or MALDI-MS analysis, because detecting more than two AGE modifications in a single molecular peptide is required (Figure 13). We believe that investigating diverse and multiple AGE patterns is beneficial for research and can help explain changes in protein function (e.g., dysfunction, activation) [49,51].

### 7.6. Comparison of Existing Methods and Our Novel Method for the Identification and Quantification of AGEs in the Urine

In the existing methods used to analyze AGEs in the urine, free-type AGEs (e.g., pentosidine, CML, and CEL) can be obtained from AGE-modified proteins treated with acid hydrolysis, which are then identified and quantified [36,37,40,55]. Although the relationship between free-type AGEs and diseases can be evaluated, the original proteins from which modified AGEs are derived cannot be identified. In contrast, our novel method can detect individual AGE-modified proteins using ESI-MS/MALDI-MS. Kato et al. performed a competitive ELISA technique to quantify pentosidine-modified proteins [65], however, using our approach it is possible to identify and quantify individual AGE-modified proteins using slot blot analysis and anti-AGE antibodies, in addition to competitive ELISA. Moreover, the slot blot analysis in our method will avoid various preprocessing steps used in existing methods (e.g., concentration of samples or acetone precipitation of to collect proteins) which are required for competitive ELISA.

### 7.7. Advantages of Using Three Methods to Analyze AGEs

Quantifying free-type AGEs may contribute to biomarker research because the structure of free-type AGEs, rather than AGE-modified proteins, may be associated with individual diseases or dietary lifestyles [44,61]. If specific AGE-modified proteins are related to diseases and have potential as biomarkers, their identification should be performed by slot blotting and ESI-MS or MALDI-MS. Although the structure of each AGE-modified protein should ideally be determined, slot blot analysis can detect AGEs with anti-AGE antibodies even if the structure is unclear. Conversely, understanding the effects of diverse or multiple AGE patterns requires revealing their structures. These three methods complement each other’s weaknesses.

### 7.8. Desirable Practical Clinical Applications

We believe that our novel method will be beneficial in clinical practice, as free-type AGEs and AGE-modified proteins may serve as biomarkers for different diseases (especially for LSRD-associated AGEs in T2DM, NASH, and CVD), and dietary AGEs can be detected and quantified. The novel method can assess different AGEs in urine compared with the previous methods used for the detection and quantification of AGEs. However, our method requires further validation in basic and clinical studies for the detection of free-type AGEs and AGE-modified proteins in healthy controls and patients with LSRD to demonstrate their role as biomarkers before it can be applied in practical clinical applications.

### 7.9. Limitations of the Novel Strategy

In this analysis, free-type AGEs in the urine under the non-acid hydrolysis condition cannot be detected when discharged from the urethra. As a result of their small molecular weight, they do not bind to the PVDF membrane.

## 8. Conclusions

Although analyzing AGE-modified proteins in urine is challenging and requires various steps, the novel approach that combines slot blot with ESI-MS or MALDI-MS may be beneficial. This method could address previous issues and enable the identification and quantification of AGE-modified proteins, AGE-modified peptides, and free-type AGEs.

## Figures and Tables

**Figure 1 ijms-25-09632-f001:**
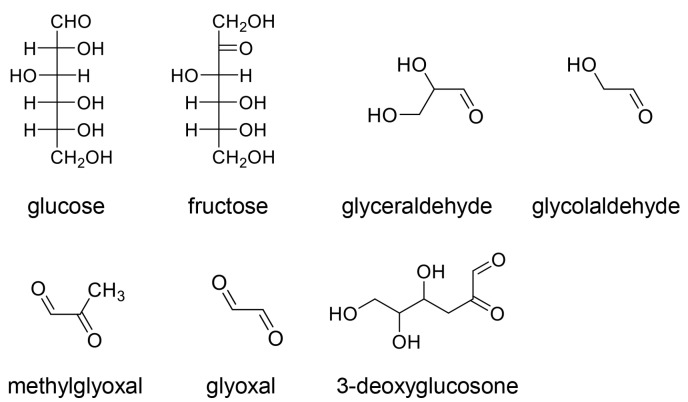
Precursor molecules of advanced glycation end products (AGEs) [38,81]. Glucose and fructose are classified as saccharides. Other compounds are produced from glucose and/or fructose. Glucose, glyceraldehyde, glycolaldehyde, methylglyoxal, glyoxal, and 3-deoxyglucosone are precursors of AGE-1, -2, -3, -4, -5, and -6, respectively.

**Figure 2 ijms-25-09632-f002:**
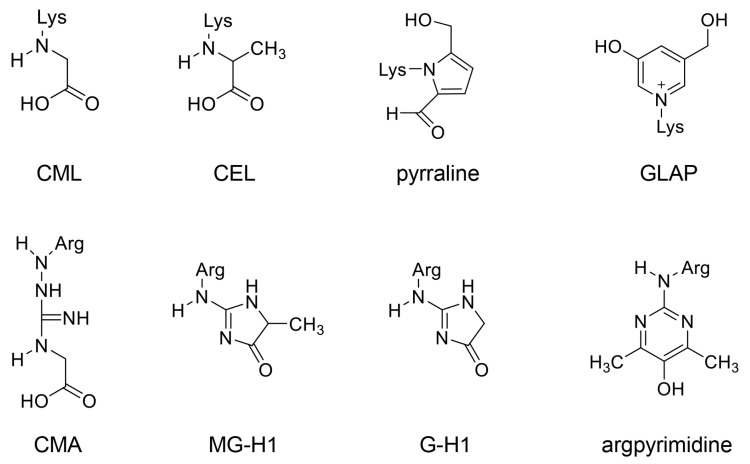
Free-type AGEs containing one amino acid residue. Lys, lysine; Arg, arginine. CML, *N*^ε^-carboxymethyl-lysine [34,48,49]; CEL, *N*^ε^-carboxyethyl-lysine [35,42,43,49]; pyrraline [49,94,95,96], GLAP, 3-hydroxy-5-hydroxymethyl-pyridinium [49,77,97]; CMA, *N*^ω^-carboxymethylarginine [48,98,99]; MG-H1, *N*^δ^-(5-hydro-5-methyl-4-imidazolone-2-yl)-ornithine (methylglyoxal-derived hydroimidazolone) [45,46,49,74,77]; G-H1, glyoxal-derived hydroimidazolone [45,49,100,101]; argpyrimidine [46,48,49,77].

**Figure 3 ijms-25-09632-f003:**
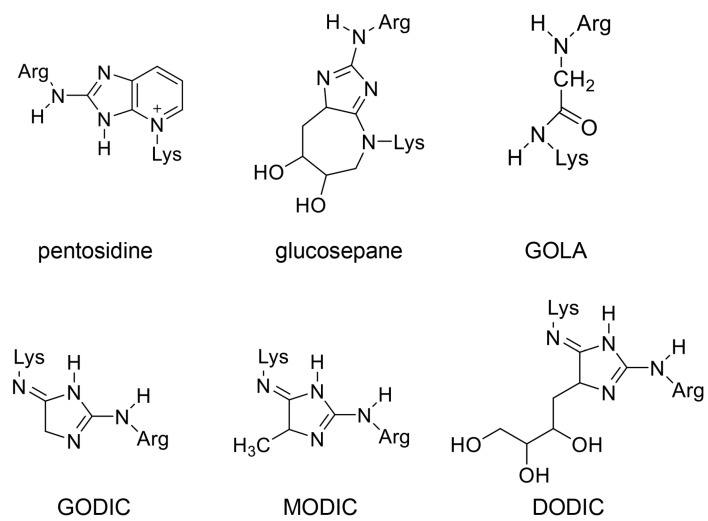
Free-type AGEs containing two amino acid residues. Lys, lysine; Arg, arginine. Pentosidine [34,35,36,37]; glucosepane [107,108,109]; *N*^ε^-{2-[(5-amino-5-carboxypentyl)-amino]-2-oxoethyl}-lysine (GOLA) [110]; glyoxal-derived imidazolium cross-link (GODIC) [111]; methylglyoxal-derived imidazolium cross-link (MODIC) [111]; 3-doxyglucosone-derived imidazolium cross-link (DODIC) [111].

**Figure 4 ijms-25-09632-f004:**
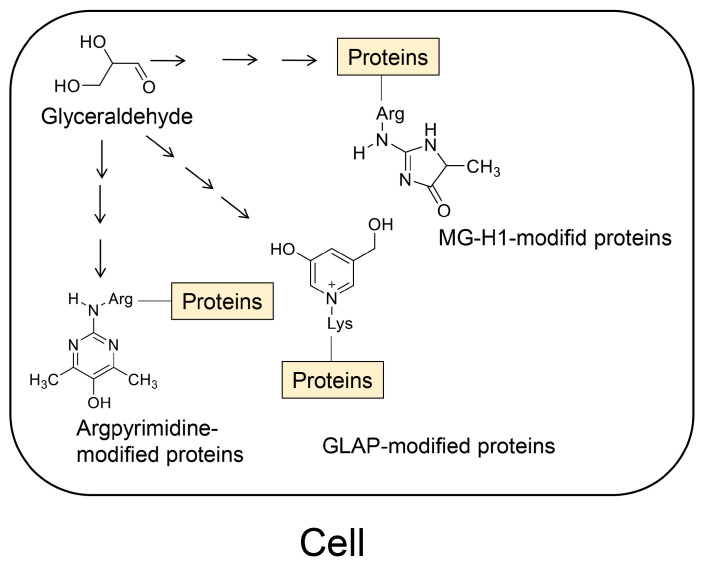
Crude advanced glycation end-product (AGE) pattern [49,51]. GLAP, MG-H1, and argpyrimidine structures were generated from glyceraldehyde, and the proteins were modified to generate each AGE in PANC-1 cells [77]. GLAP, 3-hydroxy-5-hydroxymethyl-pyridinium [49,77,97]; MG-H1, *N*^δ^-(5-hydro-5-methyl-4-imidazolone-2-yl)-ornithine (methylglyoxal-derived hydroimidazolone) [45,46,49,74,77]; argpyrimidine [46,49,77].

**Figure 5 ijms-25-09632-f005:**
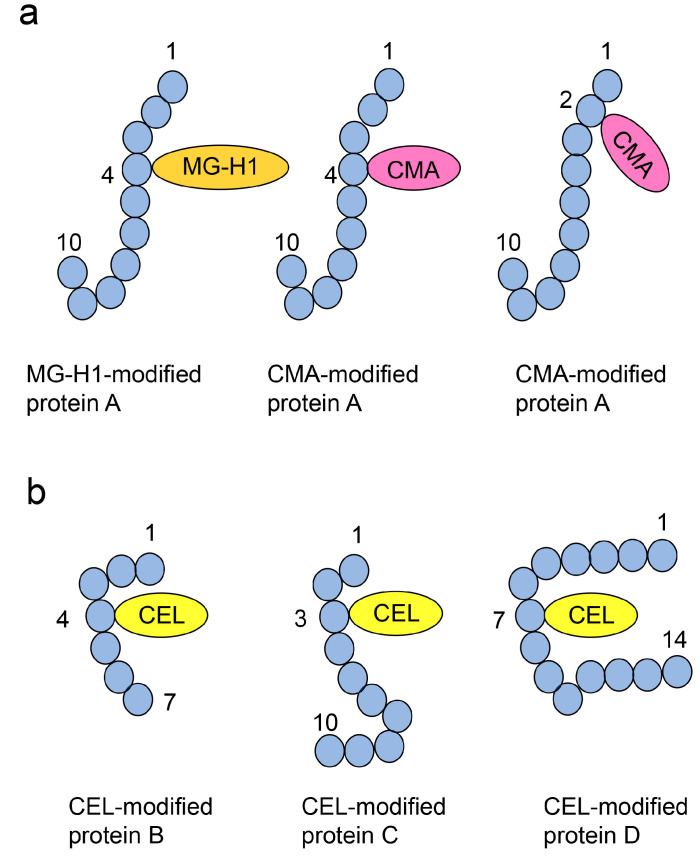
Type 1 (1A, 1B, 1C) and 2 diverse advanced glycation end-product (AGE) patterns [49,51]. Closed blue circles indicate the amino acids. The black numbers represent the residue number in proteins A, B, C, and D; CEL, *N*^ε^-carboxyethyl-lysine [35,42,43,49]. (**a**) MG-H1, *N*^δ^-(5-hydro-5-methyl-4-imidazolone-2-yl)-ornithine (methylglyoxal-derived hydroimidazolone) [45,46,49,74,77]; CMA, *N*^ω^-carboxyethyl-arginine [48,98,99]. Type 1 diverse AGE pattern. Each protein A is modified by a certain AGE at same or different amino acid, respectively. The left and middle glycated protein A are classified as Type 1A diverse AGE pattern (Protein A is modified by MG-H1 or CMA at the fourth amino acid residue). The middle and right glycated protein A are classified as Type 1B diverse AGE pattern (Protein A is modified by CMA at the fourth and second amino acid residues). The left and right glycated protein A are classified as Type 1C diverse AGE pattern (Protein A is modified by MG-H1 and CMA at the fourth and second amino acid residues, respectively). (**b**) Type 2 diverse AGE pattern. (CEL is bound to proteins B, C, and D).

**Figure 6 ijms-25-09632-f006:**
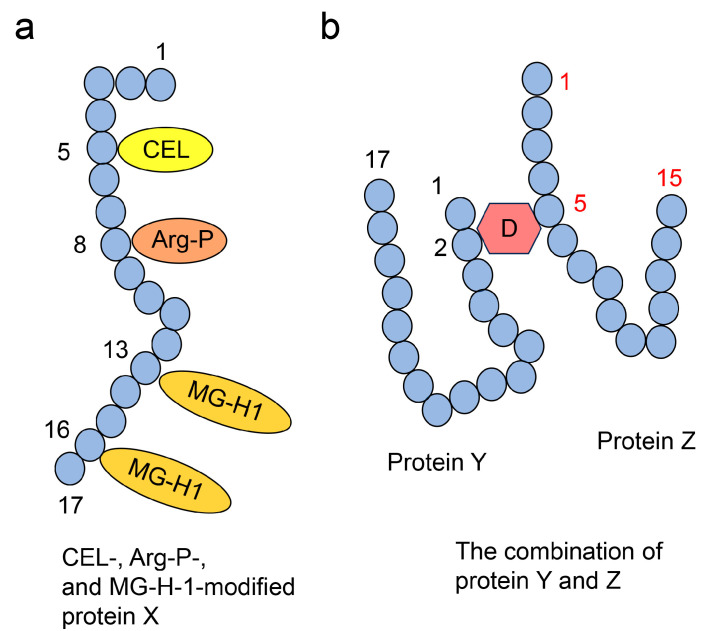
Type I and II multiple AGE patterns [49,51]. Black and red numbers indicate amino acid residue numbers. Closed blue circles represent amino acids (**a**) Type I multiple AGE pattern. CEL-, Arg-P-, and MG-H1-modified protein X, but not protein X alone (one molecule, but not one type of protein). CEL, *N*^ε^-carboxyethyl-lysine [35,42,43,49]. Arg-P: argpyrimidine [46,48,49,77]; MG-H1, *N*^δ^-(5-hydro-5-methyl-4-imidazolone-2-yl)-ornithine (methylglyoxal-derived hydroimidazolone) [45,46,49,74,77]. (**b**) Type II multiple AGE pattern showing an intermolecular covalent bond. D1: AGE structure that binds to the second amino acid residue in protein Y and the fifth amino acid residue in protein Z.

**Figure 7 ijms-25-09632-f007:**
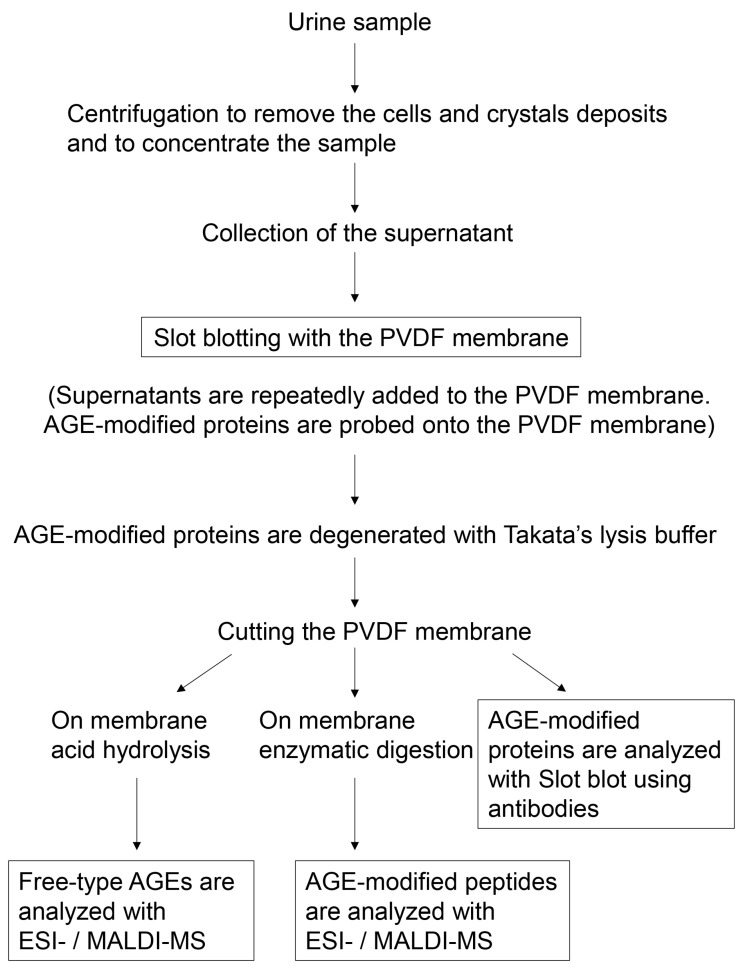
Flowchart showing the steps in the novel analysis method for AGEs in urine.

**Figure 8 ijms-25-09632-f008:**
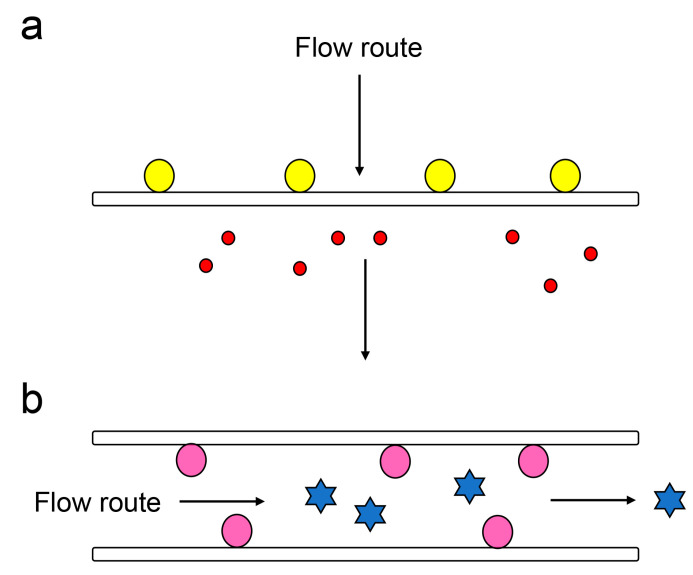
Schematic describing membrane chromatography. The open white box is the membrane. The black arrows indicate the flow path of the samples. (**a**) The sample flows perpendicular to the membrane. Closed yellow circles indicate high molecular weight compounds. Closed red circles indicate low molecular compounds. (**b**) Samples flow horizontally in the membrane. Closed pink circles indicate compounds with high affinity for the membrane. Closed blue stars indicate compounds that do not have affinity to the membrane.

**Figure 9 ijms-25-09632-f009:**
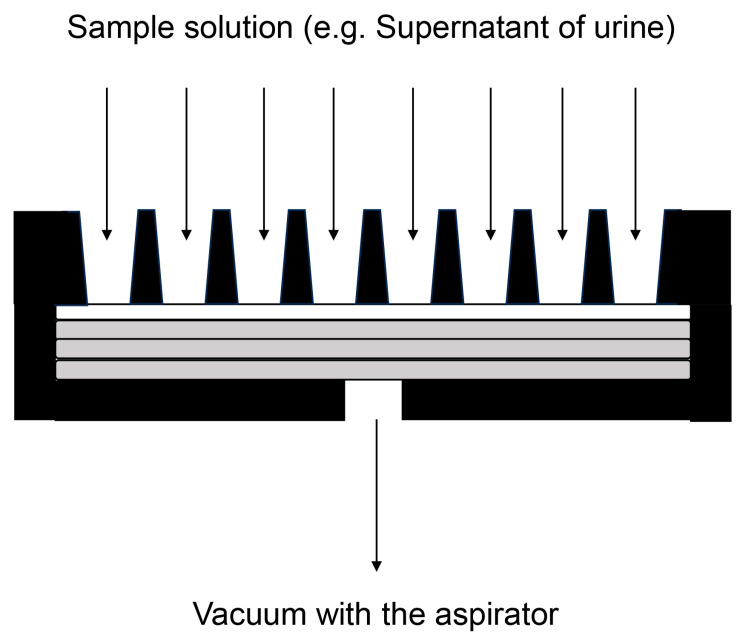
Schematic showing the application of the sample solution using a vacuum aspirator. Closed black apparatus indicates the slot blot apparatus. Closed white box indicates the PVDF membrane. Closed boxes square indicate the filter paper.

**Figure 10 ijms-25-09632-f010:**
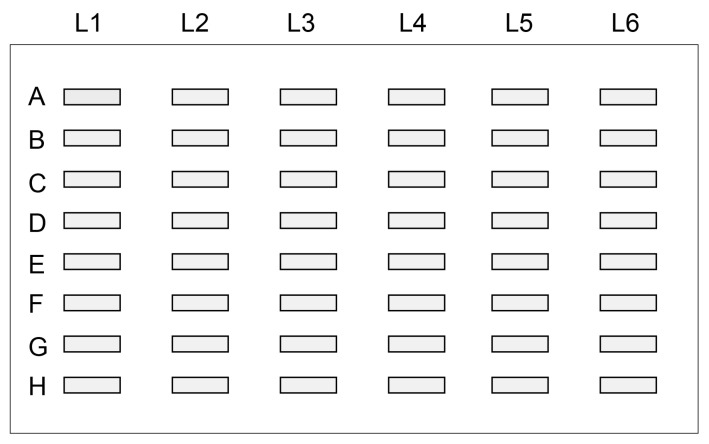
Spaces on the PVDF membrane that contain the sample solution. The closed gray boxes indicate the lanes with the sample solution. The general slot blot apparatus (e.g., Bio-dot SF microfiltration apparatus (Bio-Rad, Hercules, CA, USA)) has 48 lanes on the membrane (9 cm × 12 cm) [50,117].

**Figure 12 ijms-25-09632-f012:**
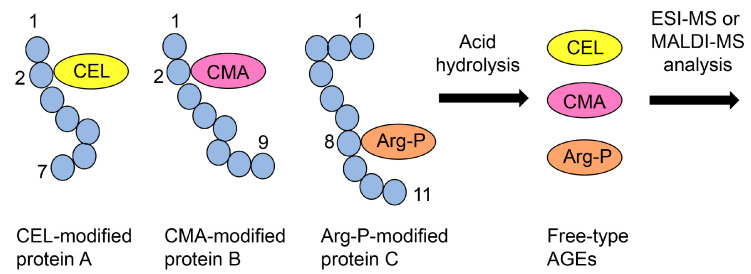
Schematic showing the ESI- or MALDI-MS analysis of free-type AGEs. CEL, *N*^ε^-carboxyethyl-lysine [35,42,43,49]; CMA, *N*^ω^-carboxyethyl-arginine [48,98,99]; Arg-P, argpyrimidine [46,48,49,77]. Closed blue circles indicate the amino acids. The black numbers represent the residue number in proteins A, B, and C.

**Figure 13 ijms-25-09632-f013:**
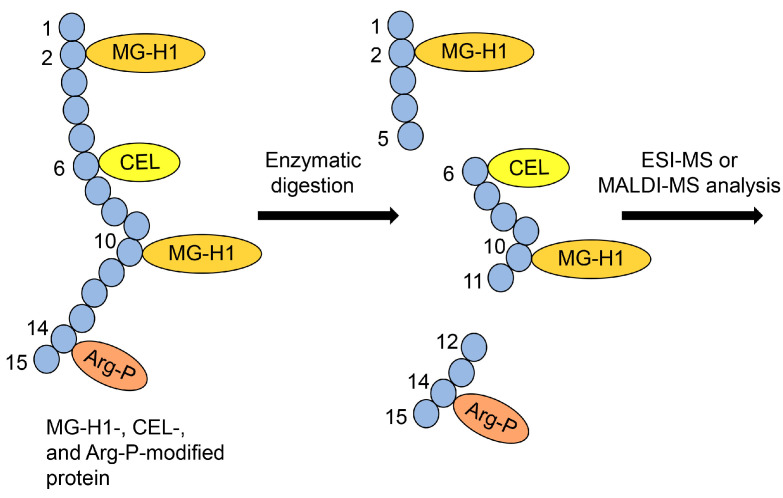
Schematic showing ESI- or MALDI-MS analysis of AGE-modified peptides. MG-H1, *N*^δ^-(5-hydro-5-methyl-4-imidazolone-2-yl)-ornithine (methylglyoxal-derived hydroimidazolone) [45,46,49,74,77]; CEL, *N*^ε^-carboxyethyl-lysine [35,42,43,49]; Arg-P, argpyrimidine [46,48,49,77]. Closed blue circles indicate the amino acids. The black numbers represent the residue number in protein.

**Table 1 ijms-25-09632-t001:** Recipe for Takata’s lysis buffer and modified Takata’s lysis buffer.

Solution A (Pre-Takata’s Lysis Buffer)	Solution B	Solution C (Takata’s Lysis Buffer)	Solution D
30 mM Tris base	1 protease inhibitor cocktail tablet/2 mL	27 mM Tris base	30 mM Tris base
7 M urea		6.3 M urea	7 M urea
2 M thiourea		1.8 M thiourea	2 M thiourea
4 *w*/*v* (%) CHAPS		3.6 *w*/*v* (%) CHAPS	4 *w*/*v* (%) CHAPS
		10 (*v*/*v*) % Solution B	4 *v*/*v* (%) Solution B
(pH 8.5)		(pH 8.5)	(pH 8.5)

**Table 2 ijms-25-09632-t002:** Previous studies have been referenced for developing solutions C and D.

Solution	References
Solution C (Takata’s lysis buffer)	[73,139,140,141,142,143,144,145]
Solution D	[146,147,148,149]

## Data Availability

The data presented in this study are available from the corresponding author upon request.

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
