# Peer review of "Slot Blot- and Electrospray Ionization–Mass Spectrometry/Matrix-Assisted Laser Desorption/Ionization–Mass Spectrometry-Based Novel Analysis Methods for the Identification and Quantification of Advanced Glycation End-Products in the Urine"

_ijms, 2024, doi:10.3390/ijms25179632_

Round 1
Reviewer 1 Report
Comments and Suggestions for Authors
This manuscript provides detailed and comprehensive comments on the novel analysis methods for identification and quantification of advanced glycation end-products in urine. However, the reviewers found that there are still some shortcomings in this manuscript that need further supplementation to be suitable for publication.
1. The author should also provide appropriate introductions to other identification and quantification methods of advanced glycation end-products in urine.
2. The author provided a detailed description of the Slot Blot and ESI-MS/MALDI-MS in the manuscript. How effective is this method in practical clinical applications? Can the author provide some examples?
Comments on the Quality of English LanguageNo problem
Author Response
Response Letter to Reviewers’ Comments
Responses to Reviewer 1
Dear Reviewer 1:
Thank you for giving us the opportunity to submit a revised draft of our manuscript titled “Slot Blot- and ESI-MS/MALDI-MS-based Novel Analysis Methods for the Identification and Quantification of Advanced Glycation End-Products in the Urine” to the International Journal of Molecular Sciences (manuscript ID: 3155442). We appreciate the time and effort the reviewers have taken to provide their valuable feedback on our manuscript; their comments have enriched the manuscript and helped us produce a more balanced account of our research. The manuscript has been reviewed by a professional English editor (Editage) to address all grammatical and syntax errors and improve the overall readability of the document.
We have provided a Graphical Abstract and inserted a new Section 6 to introduce the previous technologies for the identification and quantification of AGEs in urine.
Comments and Suggestions for Authors
This manuscript provides detailed and comprehensive comments on the novel analysis methods for identification and quantification of advanced glycation end-products in urine. However, the reviewers found that there are still some shortcomings in this manuscript that need further supplementation to be suitable for publication.
Comment 1: The author should also provide appropriate introductions to other identification and quantification methods of advanced glycation end-products in urine.
Response 1: We have inserted a new section 6 (Title: Previous Methods used for the Identification and Quantification of AGEs in the Urine) to introduce the previous technologies for the identification and quantification of AGEs (pentosidine, CML, and CEL) in the urine.
The information about pentosidine has been described on the basis of the reports in Ref. 34–37, 48, 64, and 65. The information regarding CML and CEL has been described based on the reports in Ref. 40 and 55. These sentences are highlighted in yellow.
Comment 2: The author provided a detailed description of the Slot Blot and ESI-MS/MALDI-MS in the manuscript. How effective is this method in practical clinical applications? Can the author provide some examples?
Response 2: We believe that our novel methods will be beneficial in practical clinical applications because free-type AGEs and AGEs-modified proteins, as biomarkers for various diseases (especially LSRDs with AGEs such as T2DM, NASH, and CVD) and lifestyle of intake various dietary AGEs will be detected and quantified. These novel methods can provide multiple perspectives to assess the amounts of AGEs in the urine compared with the previous methods for the detection and quantification of AGEs. However, these methods entail collection of data regarding the use of these free-type AGEs and AGE-modified proteins as biomarkers in healthy individuals and patients with LSRD prior to their application in practical clinical use.
We have described this information in the New Section 7.7.
Comments on the Quality of English Language
No problem

Reviewer 2 Report
Comments and Suggestions for Authors
Authors proposed combined slot-blot and ESI/MALDI MS quantification of glucan metabolite in urine. Authors introduced and reviewed the current situation of the field and proposed a novel method. In general, as the opinion paper, the manuscript is well-written and interesting. But the more detailed example of the suggested method should be defined with detail, especially the method that suggested an idea as a 'quantification' method, but the detail of signal/sensitivity enhancement of the slot-blot with 'how much.' In addition with the sample preparation between ESI and MALDI should be quite different. more detail of the sample prepration and detectio method of mass spectrometry should be clarify with detail.
Author Response
Response Letter to Reviewers’ Comments
Responses to Reviewer 2
Dear Reviewer 2:
Thank you for giving us the opportunity to submit a revised draft of our manuscript titled “Slot Blot- and ESI-MS/MALDI-MS-based Novel Analysis Methods for the Identification and Quantification of Advanced Glycation End-Products in the Urine” to International Journal of Molecular Sciences (manuscript ID: 3155442). We appreciate the time and effort the reviewers have taken to provide their valuable feedback on our manuscript; their comments have enriched the manuscript and helped us produce a more balanced account of our research. The manuscript has been revised by a professional English editor (Editage) to address all grammatical and syntax errors and improve the overall readability of the document.
We have provided a Graphical Abstract and inserted a new Section 6 to introduce the previous technologies used for identification and quantification of AGEs in urine samples.
Comments and Suggestions for Authors
Authors proposed combined slot-blot and ESI/MALDI MS quantification of glucan metabolite in urine. Authors introduced and reviewed the current situation of the field and proposed a novel method. In general, as the opinion paper, the manuscript is well-written and interesting. But the more detailed example of the suggested method should be defined with detail, especially the method that suggested an idea as a 'quantification' method, but the detail of signal/sensitivity enhancement of the slot-blot with 'how much.' In addition, with the sample preparation between ESI and MALDI should be quite different. more detail of the sample prepration and detectio method of mass spectrometry should be clarify with detail.
Comment 1: The more detailed example of the suggested method should be defined with detail, especially the method that suggested an idea as a 'quantification' method, but the detail of signal/sensitivity enhancement of the slot-blot with 'how much.'
Response 1: Although Takata et al. first reported the quantification of GA-AGEs in human cell lysates using Takata’s lysis buffer in July 2017 (Ref. 73), Dr. Takanobu Takata had developed this lysis buffer in March 2014 and revealed that it enhances the signal/sensitivity in the previous laboratory. However, the data were never published and may be unsuitable as a reference for this opinion article.
Nonetheless, we have provided additional important details regarding this novel enhanced slot blot method using Takata’s lysis buffer and compared it with the previously used slot blot approach (Ref. 50). To date, no reports are available that describe the quantification of AGEs in the lysates of cultured cells/tissue, plasma, serum, saliva, urine, or other body fluids using the existing slot blot method with PVDF membranes, along with Statistical Analysis. In 2010, Takino et al. quantified GA-AGEs in human cell lysates via slot blot analysis using a commercial lysis buffer and PVDF membranes; however, they did not perform statistical analysis (i.e., no error bars were shown in the published graphs accompanying the quantification of GA-AGEs in the cell lysates) (Ref. 50). As both GA-AGEs were detected with the same anti-GA-AGEs antibody, the reason the latter study did not perform the statistical analysis was not attributable to the quality of the antibody. We believe that Takata’s lysis buffer introduced in this study favors adequate blotting and probing of proteins, including AGE-modified proteins, onto the PVDF membranes and thus will allow suitable statistical analysis. We have added this information to the new Section 7.4.2.
Comment 2: The sample preparation between ESI and MALDI should be quite different. more detail of the sample preparation and detection method of mass spectrometry should be clarify with detail.
Response 2: Several studies have described detection of peptides following enzymatic digestion using MS and the relevant reference database. These peptides require reduction of the disulfide bond and alkylation of the thiol group and detection using either ESI or MALDI-MS (Refs. 154,155). However, the ionization step is different for the two methods; ESI-MS can perform ionization without additional reagents, whereas the sample preparation for MALDI-MS requires mixing with a matrix reagent for ionization. In addition, multiple charged peptide ions are easily produced in ESI-MS compared with those in MALDI-MS. In contrast, only single charged peptide ions are easily produced in MALDI-MS (Refs. 45–47, 154, 155). As the retention times of peptides (AGE-modified peptide) can be selectively determined, their quantification is possible based on the HPLC-ESI-MS analysis (Ref. 154,155). We have added this information in the new Section 7.5.1 in the revised manuscript.

Round 2
Reviewer 2 Report
Comments and Suggestions for Authors
The authors revised the major insufficient points. Even the paper is an Opinion and lacks a comprehensive presentation of the current method and novel method. After the first revision, the paper still needed some minor corrections, details of which are given below.
1. The introduction is too long and does not focus on the main topic. The introduction should clearly articulate the gaps in current methodologies that this study aims to address.
2. A more precise description should be clarified for the suggested methodology part (related to the Figure 7 workflow).
3. This paper claims a novel method combining slot blotting and ESI-MS/MALDI-MS for the detection of AGEs in urine but does not adequately compare this approach with existing methods. This point should be necessary to add.
Author Response
Responses Reviewer’s Comments (Round 2)
Responses to Reviewer 2
Dear Reviewer 2:
Thank you for giving us the opportunity to submit a revised draft of our manuscript titled “Slot Blot- and ESI-MS/MALDI-MS-based Novel Analysis Methods for the Identification and Quantification of Advanced Glycation End-Products in the Urine” to the International Journal of Molecular Sciences (manuscript ID: 3155442). We appreciate the time and effort taken to provide valuable feedback on our manuscript; their comments have enriched the manuscript further and have helped us produce a more balanced account of our research. The manuscript has been reviewed by a professional English editor (Editage) to address all grammatical and syntax errors and improve the overall readability of the document.
Comments and Suggestions for Authors
The authors revised the major insufficient points. Even the paper is an Opinion and lacks a comprehensive presentation of the current method and novel method. After the first revision, the paper still needed some minor corrections, details of which are given below.
Comment 1: The introduction is too long and does not focus on the main topic. The introduction should clearly articulate the gaps in current methodologies that this study aims to address
Response: We have modified the sentences regarding “blood, urine, and saliva markers” used for various diseases in the Introduction section. (Yellow highlighted text)
“We believe that urine is a suitable sample to study biomarkers of diseases or disorders given the high amounts of sample (e.g., 50–150 mL) that can be collected painlessly from the patient.”
We have inserted the above sentence in the middle of the Introduction section (Pink highlighted text).
“Furthermore, our novel method can avoid the need for pre-processing of urine required by existing methods and can perform multiple analyses of free-type of AGEs and AGEs-modified proteins.”
We have inserted this sentence in the last part of the Introduction section. (Pink highlighted text).
Comment 2: A more precise description should be clarified for the suggested methodology part (related to the Figure 7 workflow).
Response: In our novel method, urine samples can be concentrated using existing methods; however, even if the concentrations obtained are lower, supernatants can be repeatedly collected and deposited onto the PVDF membrane.
“In this step, researchers can use existing methods to concentrate urine [66–69], however, if the concentration obtained is lower, the collected supernatants can be repeatedly deposited onto the PVDF membrane [117]. In contrast, acetone precipitation not to be performed using our novel method as this step is performed in existing methods to collect proteins rather than to concentrate the sample [69]. The collected supernatant is then deposited into the lanes of the slot blot apparatus equipped with a PVDF membrane. This step should be performed repeatedly because this will allow to further increase the concentration of sample that is deposited onto the PVDF membrane. After a sufficient amount of concentrated supernatant is deposited, AGE-modified proteins are bound to the PVDF membrane [50,73].”
We have added this description to Section 7.2. (Yellow highlighted text).
More, we revised the Figure 7.
Comment 3: This paper claims a novel method combining slot blotting and ESI-MS/MALDI-MS for the detection of AGEs in urine but does not adequately compare this approach with existing methods. This point should be necessary to add.
Response: To explain the differences between existing methods and our proposed method, we have inserted a new section 7.6. (Yellow highlighted text).
“7.6. Comparison of Existing Methods and Our Novel Method for the Identification and Quantification of AGEs in the Urine
In existing methods used to analyze AGEs in the urine, free-type AGEs (e.g., pentosidine, CML, and CEL) can be obtained from AGE-modified proteins treated with acid hydrolysis, which are then identified and quantified [36,37,40,55]. Although the relationship between free-type AGEs and diseases can be evaluated, the original proteins from which modified AGEs are derived cannot be identified. In contrast, our novel method can detect individual AGE-modified proteins using ESI-MS/MALDI-MS. Kato et al. performed a competitive ELISA technique to quantify pentosidine-modified proteins [65], however, using our approach it is possible to identify and quantify individual AGE-modified proteins using slot blot analysis and anti-AGE antibodies, in addition to competitive ELISA. Moreover, the slot blot analysis in our method will avoid various preprocessing steps used in existing methods (e.g., concentration of samples or acetone precipitation of to collect proteins) which are required for competitive ELISA.”
